# High-Calorie Food-Cues Impair Conflict Control: EEG Evidence from a Food-Related Stroop Task

**DOI:** 10.3390/nu14214593

**Published:** 2022-11-01

**Authors:** Yong Liu, Jia Zhao, Yizhou Zhou, Ruiyu Yang, Beichen Han, Yufei Zhao, Yazhi Pang, Hong Yuan, Hong Chen

**Affiliations:** 1Key Laboratory of Cognition and Personality (Ministry of Education), School of Psychology, Southwest University, Chongqing 400715, China; 2School of Psychology, Southwest University, Chongqing 400715, China; 3School of Education, Chongqing Normal University, Chongqing 401331, China

**Keywords:** conflict control, high-calorie foods, food-related Stroop task, N2 and P3, theta power

## Abstract

Long-term excessive intake of high-calorie foods might lead to cognitive impairments and overweight or obesity. The current study aimed to examine the effects of high-calorie foods on the behavioral and neurological correlates of food-related conflict control ability. A food-related Stroop task, which asked the participants to respond to the food images and ignore the calorie information, were employed. A total of 61 individuals were recruited and who completed the food-related Stroop task with event-related potentials (ERPs). Participants exhibited a slower reaction time and lower accuracy in high-calorie food stimuli than that in low-calorie food stimuli. The ERP results exhibited a reduction in N2 amplitudes when responding to high-calorie food stimuli compared to when responding to low-calorie food stimuli. In addition, time-frequency analysis revealed that theta power induced by low-calorie food stimuli was significantly greater than that of high-calorie food stimuli. The findings indicated that high-calorie foods impair food-related conflict control. The present study expands on the previous studies of the neural correlates of food cues and provides new insights into the processing and resolving of conflicting information for eating behavior and weight control.

## 1. Introduction

We live in an environment filled with various food temptations at our fingertips and we are constantly struggling between maintaining a healthy eating style and giving in to immediate food temptations. The excessive intake of high-calorie foods is considered a contributing factor to obesity, while resisting high-calorie foods is considered to require successful executive function [1,2,3,4]. Being overweight and obese can, in turn, have negative effects on executive functions including response inhibition, conflict control, cognitive flexibility, and decision-making [5,6,7,8,9,10,11].

Conflict control refers to the information processing capability of the brain to monitor and respond to the emergence of conflicts and is considered a core component of the brain’s executive function [12]. Previous behavioral data indicate that unhealthy food (high-calorie food) may have negative effects on individuals’ conflict control. Nijs and colleagues (2010) aims to investigate the food-related conflict control in individuals with obesity. During a food-related Stroop task, participants were instructed to respond as fast as possible to the font color of the words [grouped as food-related words (such as chocolate, meatball) and neutral office-related words (such as scissor, desk)] shown on the screen and ignore the meaning of the words. They found greater reaction times to food-related words than to neutral words in participants of both obese and normal-weight groups, which indicated general attentional biases toward food [13]. Lyu et al. used the food-related flanker task to investigate the conflict control ability in Chinese females who report binge eating disorder characteristics and found that while the binge eating group displayed a slower reaction time to incongruent trials than that of congruent trials, the control group did not exhibit such a pattern, which indicated that the binge eating group had a relative conflict control ability deficit [14]. Smith et al. utilized a food-choice task to explore conflict control when making decisions between healthy and unhealthy food. All participants were asked to choose the healthier food during the task. They found that more self-reported cravings and desires for unhealthy food predicted a greater response conflict. The authors believed that greater conflict management would be required for individuals to complete the food-choice task efficiently [15]. In addition, during a food choice task, individuals who scored higher on restrained eating were more likely to choose healthier foods compared to high-calorie foods and exhibited less self-control conflict when choosing healthier foods compared to individuals who scored lower on restrained eating [1]. Zhang et al. (2021) investigated the effects of food-related thought suppression on conflict control ability in restrained-eating adults. They found that after inhibiting thoughts about eating, the restrained eaters chose more high-calorie foods. The findings also indicated that the suppression of food-related thoughts could lead to a reduction in the ability to monitor conflicts between current behaviors and goals, which might lead to unhealthy eating behaviors [16]. In addition, high-calorie food may have a negative effect on the other cores of executive function. For example, Meule et al. explored the effects of high-calorie food on working memory in high food cravers and low food cravers. All participants performed an n-back task (the stimuli consisted of high-calorie food and neutral items). They found slower reaction times and lower accuracy in response to high-calorie food than to neutral items regardless of reported cravings, which indicated that high-calorie food cues have immediate negative effects on working memory performance [17].

Adding to behavior studies, prior neurophysiology studies have also investigated the relationships between inhibition control and food-cues. Although most of the previous studies have shown that high-calorie foods have a wide range of effects on P2, N2, P3, N450, and late positive potential (LPP) amplitudes, they have revealed mixed findings. Asmaro and his colleagues compared the neural-difference induced by high-calorie foods (e.g., chocolate, milk.) and neutral objects (e.g., chair) during discrimination tasks using event-related potentials (ERPs, a neuroscience tool with high temporal resolution, which is often used in cognitive psychology research), and found that high-calorie foods elicited significant enhancement of P2 and LPP amplitudes. The findings indicated that participants exhibited attentional bias by allocating more attentional resources to high calorie food stimuli compared to general stimuli [18]. When asked to inhibit during a go/no-go or oddball task, the participants exhibited larger N2 amplitudes when inhibiting to high-calorie food compared to when inhibiting to low-calorie or non-food-related stimuli [19,20,21,22]. Liu et al. found greater N2b amplitudes in response to high-calorie food cues than those in response to low-calorie food cues during a food-related go/no-go task [7]. In addition, Carbine et al., found that during a food-related go/no-go task, the P3 amplitudes were greater when inhibiting to high-calorie food cues compared to when inhibiting to low-calorie food cues [23]. Franken et al. also found that sweet food elicited greater P3 amplitudes compared to neutral-flavored food [24]. However, in successful restrained eaters, Kong et al. found greater P3 amplitudes to neural image cues and low-calorie food cues than to high-calorie food cues [21]. However, using a go/no-go task, Lapenta et al. (2014) found no difference in N2 and P3 amplitudes when inhibiting toward food cues or furniture cues [25]. Nijs and colleagues (2010) set out to investigate food-related conflict control abilities in individuals with obesity. During a food-related words Stroop task, the participants were instructed to respond as fast as possible to the font color of the word shown on the screen and ignore the meaning of the word. They found enhanced P2 amplitudes in the obese group, which indicated that the obese participants’ attentional bias towards food cues occurred at a heighted automatic, habitual, and preconscious level [13]. A study conducted by Liu et al. showed that there were decreases in N2 and N450 amplitudes and increases in P3 and LPP amplitudes in a food-related conflict task in individuals who were overweight [6]. Woltering and colleagues used an attention blink task to reveal decreased P3 amplitudes during the onset of food stimuli in obese adolescents compared to their normal-weight counterparts. Lower P3 amplitudes were related to a higher body mass index (BMI) assessment. The findings suggested that an automatic attentional bias to food should be considered as an key factor in the process of tackling the rising obesity crisis [26]. As the findings present, the existing mechanism of high-calorie food on conflict control ability remains unclear. It is beneficial to explore how high-calorie food cues influence executive function since it would provide insights into the challenges of developing healthier eating behaviors.

Theta power has also been a neural marker of focus for previous studies when discussing conflict control ability regarding food-related stimuli. Theta power of the medial frontal cortex is sensitive to perception conflicts and reaction conflicts. Theta band activities are usually involved in conflict processing, especially in the neural network that connects reaction conflicts [27,28]. In addition, frontal midline theta power is related to increased cognitive control [29,30,31]. A previous study aimed at examining the neural markers that are in support of the effectiveness of food-related go/no-go training on reducing food intake found that food cues requiring withholding responses induced larger increases in theta power compared to cues that do not relate to withholding responses [32].

Up to now, there have been few EEG studies that have utilized a food-related Stroop task to investigate the effects of high-calorie foods on conflict control. To add to previous studies, the present study examined the influence of high-calorie food cues on the neural correlates of conflict control. Based on previous studies, the neural correlates discussed in the present study are N2, P3, and theta power and we hypothesized that high-calorie foods would impair conflict control ability, which would be reflected by a slower reaction time, lower accuracy, a reduction in N2 amplitudes and theta power, as well as an increase in P3 amplitudes when reacting to high-calorie foods compared to when reacting to low-calorie foods during the food-related Stroop task.

## 2. Methods

### 2.1. Participants

Participants (*n* = 61; 29 females; M_age_ = 20.25 years, SD_age_ = 1.96; M_BMI_ = 21.75 kg/m^2^, SD_BMI_ = 2.98) were recruited from Chongqing Normal University, Chongqing. Before participating in the study, all participants were required to abstain from substances (including tea and coffee) and medications that could potentially affect their concentration during the study, and were required to disclose any history of psychological disorders. They were also required to have normal or corrected-to-normal vision. The participants were given the chance to ask any questions about their participation before consenting to the study. The present study was approved by the Southwest University Ethics Committee.

### 2.2. Food-Related Stroop Task

The food-related Stroop task (Figure 1) employed in the current study was modified from the previous version used by Liu et al. [6]. The stimuli were food pictures (high- and low-calorie foods) and calorie information text (high-calorie or low-calorie) attached to them. Congruent and incongruent trials were generated by combining different stimuli. A congruent trial would be when the displayed food cue with text was consistent, such as a combination of a high-calorie food cue and the words “high-calorie”, and vice versa for incongruent trials. Participants were instructed to respond to only the food pictures and ignore the words (calorie information). During the food-related Stroop task, the participants were instructed to press the “F” (or “J”) key on the keyboard if the high-calorie foods were displayed and to press the “J” (or “F”) key if the low-calorie foods were displayed. The responses were counterbalanced across the participants in the study. In the task, the trial proceeded as follows: after a fixation appeared on the screen for 500 ms, the stimuli would be presented on the monitor until the participants responded or would disappear automatically after 2000 ms of first appearing. The stimuli were then followed by an inter-stimuli interval of 1000 ms. The food-related Stroop task consisted of a practice block of 20 trials and an experimental block of 160 trials. Only data from the experimental trials were used in the analysis process. The participants were instructed to sit as still as possible and to minimize eye blinking during the task in order to reduce potential experimental artifacts of the EEG data collection.

### 2.3. Behavioral Analysis

Two 2 (food: high- and low-calorie food) × 2 (congruence: congruent and incongruent) repeated measures ANOVA was conducted on the reaction time (RT) and accuracy (ACC). All analyses were conducted by SPSS 25.0. The *p*-values were computed for deviation in all analyses, based on the Greenhouse–Geisser method. Post-hoc t-tests were conducted with Bonferroni correction for multiple pairwise comparisons.

### 2.4. EEG Recording and Analyses

EEG data were recorded from 64 scalp sites using tin electrodes mounted in an elastic cap (brain products GmbH, Gilching, Germany), with the reference electrode placed on the fronto-central aspect and a ground electrode on the medial frontal aspect. All inter-electrode impedance was maintained below 5 KΩ.

EEG data preprocessing was performed via EEGLAB, an open-source toolbox available on the Matlab software. Individual and grand ERP averages were created for the food-related Stroop task. We first down sampled the data from 1000 Hz to 256 Hz and performed high-pass filtering at 0.1 Hz and low-pass filtering at 45 Hz. The mean values of the left and right mastoids were selected as the re-reference. Data were epoched from 200 ms prior to stimuli onset to 1000 ms after the onset, and were baseline corrected to the pre-stimuli interval. Trials with large fluctuations in amplitudes were removed before the independent component analysis (ICA). The components, including EOG artifacts (ocular movements and eye blinks) and head movement, were then removed from the results of the ICA. Based on the topographical distribution of the grand-average ERP activities, the ERP components and their time window were as follows: N2 (250–360 ms), and P3 (360–400 ms). Based on previous studies, the following electrode sites were selected, Fz, FCz, Cz, CPz, and Pz. Two 2 (food: high- and low-calorie food) × 2 (congruence: congruent and incongruent) × 5 (electrode site: Fz, FCz, Cz, CPz, and Pz) repeated measures ANOVAs were conducted on the mean amplitudes of N2 and P3.

The time-frequency analysis for the EEG data, employed a windowed Fourier transform (WFT) with a fixed 250 ms width Hanning window. The WFT yielded a complex time-frequency spectral estimate *F (t, f)* at each point *(t, f)* of the time-frequency plane extending from −200 ms to 1000 ms in the time domain, and from 1 Hz to 30 Hz (in step of 1 Hz) in the frequency domain, for each single trial. A baseline correction was applied at the subject level using the pre-stimulus interval (pre-stimulus −200 to 0 ms) to calculate the change of power according to the formula:TFD(t,f)=P(t,f)−R(f)
where $P(t,f)={\left|F(t,f)\right|}^{2}$ is the power spectral density at a given time-frequency point *(t, f)*, and *R(f)* is the averaged power spectral density of the signal enclosed within the pre-stimulus reference interval (−200 to 0 ms before the onset of the stimulation) for each estimated frequency *f*. Brain rhythmic activity of the theta (4–8 Hz, 80–250 ms) was selected in the current analysis. A 2 (food: high- and low-calorie food) × 2 (congruence: congruent and incongruent) × 5 (electrode site: Fz, FCz, Cz, CPz, and Pz) repeated measures ANOVA was conducted on the value of theta.

All analyses were conducted via SPSS 25.0. Based on the Greenhouse–Geisser method, *p*-values were computed for deviation in all analyses. Post-hoc t-tests were conducted with Bonferroni correction for multiple pairwise comparisons.

## 3. Results

### 3.1. Behavior Results

The results on RT (Figure 2A) showed a main effect of food, *F* (1, 60) = 39.30, *p* < 0.001, partial η^2^ = 0.40, RT to high-calorie foods was significantly greater than that to low-calorie foods. The results on RT also showed a main effect of congruence, *F* (1, 60) = 39.30, *p* = 0.001, partial η^2^ = 0.16, RT in incongruent trials was significantly greater than that in congruent trials.

The results on ACC (Figure 2B) showed a main effect of food, *F* (1, 60) = 19.48, *p* < 0.001, partial η^2^ = 0.25, ACC of high-calorie foods was significantly lower than that of low-calorie foods. The results on ACC also showed a main effect of congruence, *F* (1, 60) = 5.01, *p* = 0.03, partial η^2^ = 0.08, ACC in congruent trials was significantly greater than that in incongruent trials.

### 3.2. EEG Results

Grand average ERPs for N2 and P3 at Fz and topography plots are shown in Figure 3A. The theta power at Fz is shown in Figure 3B.

N2

The results on N2 amplitudes showed an interaction of food and electrode site, *F* (4, 240) = 13.55, *p* < 0.001, partial η^2^ = 0.18. The simple effect analysis showed that N2 amplitudes of low-calorie foods were significantly greater than those of high-calorie foods at Fz, *F* (1, 60) = 4.97, *p* = 0.03, partial η^2^ = 0.08. The results also showed an interaction of congruence and electrode site, *F* (4, 240) = 3.18, *p* = 0.048, partial η^2^ = 0.05. The simple effect analysis showed that N2 amplitudes in incongruent trials were significantly greater than those in congruent trials at Fz [*F* (1, 60) = 6.49, *p* = 0.013, partial η^2^ = 0.10], FCz [*F* (1, 60) = 5.41, *p* = 0.023, partial η^2^ = 0.08], and Cz [*F* (1, 60) = 7.09, *p* = 0.01, partial η^2^ = 0.11].

P3

The results on P3 amplitudes showed a main effect of congruence, *F* (1, 60) = 24.55, *p* < 0.001, partial η^2^ = 0.29, P3 amplitudes in congruent trials were significantly greater than those in incongruent trials.

Theta

The results on theta showed a main effect of food, *F* (1, 60) = 8.25, *p* = 0.006, partial η^2^ = 0.12, theta power of low-calorie foods (0.77) was significantly greater than that of high-calorie foods (0.59). The results also showed an interaction of congruence and electrode site, *F* (4, 240) = 9.46, *p* = 0.001, partial η^2^ = 0.14. The simple effect analysis showed that theta power in incongruent trials was significantly greater than that in congruent trials at Pz, *F* (1, 60) = 4.15, *p* = 0.046, partial η^2^ = 0.07.

## 4. Discussion

The present study used an originally modified food-related Stroop task to explore the effects of high-calorie foods on conflict control and the underlying neurophysiological responses. The hypothesis was partially confirmed by our original method of examination. The results showed that the RT to high-calorie food cues was greater than that to low-calorie food cues, whereas the ACC of high-calorie food cues was significantly lower than that of low-calorie food cues. We also found that high-calorie foods induced lower N2 amplitudes and theta power compared to low-calorie foods. No significant difference in P3 amplitudes were found between high-calorie and low-calorie food cues.

As our modified task elicited data from both the mental processing of food images combined with calorie information, as well as motor response performance, the current results support previous findings that high-calorie foods impair cognitive performance [17]. Adding to previous studies, the current study employed a novel paradigm in which the direct influences of food-related stimuli on reaction time and accuracy could be measured. A previous study showed that individuals with binge eating disorder exhibited a flanker effect to food-related stimuli while the controls did not [14], which might indicate that binge eating impairs food-related conflict control. It is also found that higher reported cravings for unhealthy foods (high-calorie foods) are related to greater conflict when making decisions between healthy and unhealthy foods [15], thus, indicating that high-calorie foods might have a negative effect on conflict control. Consistent with a previous study using an n-back task and food related flanker task [17,33], the current finding indicates that high-calorie foods effect individuals’ conflict control, which was reflected by slower reaction time and lower accuracy. Longer reaction time in reaction to high-calorie foods might suggest that more attentional efforts are required when responding to high-calorie foods compared to low-calorie foods, thus, resulting in longer processing time [33,34]. According to the dual-process theories of self-control, a healthy lifestyle is depended on the balance between impulsive and reflective systems, while the excessive consumption of high-calorie foods indicate that this cognitive balance is not sufficiently maintained [3,35,36]. In return, deficiencies in the ability to monitor high-calorie food conflicts might also lead to overeating or obesity. Obese individuals displayed marked attentional bias to high-calorie foods and food-related stimuli compared to lean individuals [37]. Enhanced spatial memory for high-calorie foods also significantly predicted a higher BMI [38,39]. Therefore, existing evidence indicates that conflict control ability plays a beneficial role in adopting healthier eating behaviors.

The present study showed a reduction in N2 amplitudes in high-calorie foods compared to low-calorie foods. Depending on the stimuli and tasks displayed, N2 amplitudes reflect sensory processing, response inhibition, and conflict monitoring [20,40]. Folstein and Van concluded that the fronto-central N2 is relevant to cognitive control, which included response inhibition, response conflict, and error monitoring [40]. In addition, N2 reflects the detection and monitoring of conflicts when confronted with choosing between correct and incorrect options [40,41]. Previous studies showed that obese and overweight individuals exhibited decreased N2 amplitudes compared to normal-weight individuals [5,6,7]. Decreased N2 also indicates a deficit in the ability to recruit neural resources [42]. Thus, the reduced N2 amplitudes in reaction to high-calorie foods would indicate that the individuals are less able or less effective when allocating attentional and neural resources to monitor food-specific conflicts when encountering high-calorie foods.

The result of decreased theta power in high-calorie foods was consistent with the result of reduced N2 amplitudes in high-calorie foods. Frontal midline theta is located in the anterior cingulate cortex (ACC) and the auxiliary frontal motor area of the medial frontal cortex (MFC) [29] and is part of the physiological mechanism that operates cognitive control through a series of coupling mechanisms [43]. Greater frontal midline theta power is found to be related to increased cognitive control [29,30,31]. The significant increase in theta-band oscillation of the frontal midline is related to an increased capability for adaptation, behavioral regulation and overcoming conflicts [44,45]. Previous studies have found that theta initiate a greater increase when experiencing stimuli such as stimulus-response conflicts and response errors [27,46]. Therefore, the theta power is significantly lower when reacting to high-calorie foods compared to when reacting to low-calorie foods, which might indicate that high-calorie foods affect individuals’ ability for conflict monitoring, regulation, and/or adaptation.

However, we did not find a difference in P3 amplitudes between high-calorie foods and low-calorie foods. P3 reflects various cognitive processes including working memory and attention allocation [47]. A previous study has shown that the onset of food-related cues elicited a significant increase in P3 amplitudes, which would suggest that the salience of food-related cues influence cognitive control [22]. Food-related stimuli elicited greater P3 amplitudes than neutral stimuli [22], and high-calorie foods elicited greater P3 amplitudes compared to low-calorie foods [23], which would indicate that more cognitive resources were required when responding to food-related stimuli, especially high-calorie foods. Therefore, we hypothesized that high-calorie foods would affect the individuals’ cognitive processing, which would be reflected by enhanced P3 amplitudes. The slower reaction time and lower accuracy of the current study supported the hypothesis that more cognitive resources and efforts are needed when reacting to high-calorie foods compared to when reacting to low-calorie foods. However, no significant difference in P3 amplitudes between high-calorie and low-calorie foods was found in the present study. We believe a reason might account for the finding. P3 represents a late motor inhibition process, which occurs when a decision to withhold a response is made [48,49,50]. The food-related Stroop task did not ask participants to withhold their response. Therefore, high-calorie foods did not have effects on the individual’s response inhibition ability during the food-related Stroop task.

The present study has several limitations. First, the participants’ food cravings were not measured. The relationship between food craving and food-related conflict control was therefore not explored in the current study. Future research could measure the state of the food craving (e.g., food craving questionnaire state) before and after the food-related Stroop task. Second, the current study focused on the temporal measures of brain activity of the EEG data, hence the lack of spatial measures of brain activity. Future studies could explore both the temporal and spatial neural indices in a food-related Stroop task by collecting both EEG and fMRI data. Third, general stimuli (e.g., flowers or office supplies) were not adopted in the current study, and the influence of other characteristics such as the stimuli colors or nutrient density on the experimental findings was not considered in the present study, which might be worth exploring in future studies.

## 5. Conclusions

In conclusion, this study uses an originally modified food-specific Stroop task and EEG to complement previous literatures. The current original version of the food-related Stroop task may be a promising instrument to be implicated in future brain imaging studies to further demonstrate the nuances of various types of foods on cognitive processing. Furthermore, the task could be adopted to explore the underpinnings of the relationship between high-calorie foods and conflict control, including its behavioral consequences. The study has shown that high-calorie foods impair food-related conflict control ability, expanding the studies of the neural mechanism of food cues while providing new insights into the processing and solving of conflicting information for healthier eating behavior and weight control.

## Figures and Tables

**Figure 1 nutrients-14-04593-f001:**
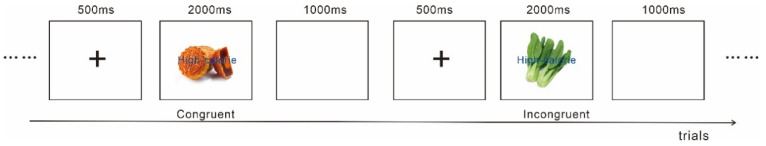
Two trials from the food-related Stroop task.

**Figure 2 nutrients-14-04593-f002:**
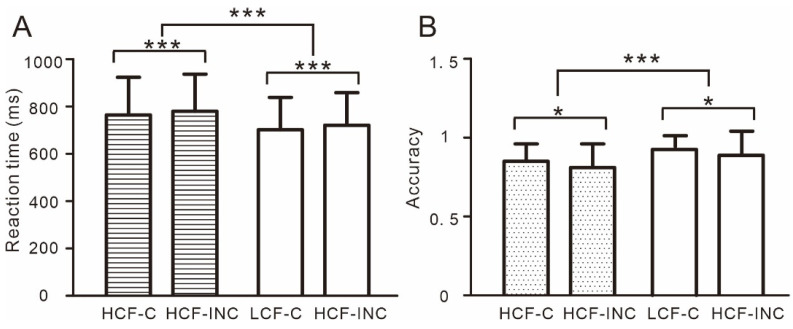
The behavioral results ((**A**). reaction time and (**B**). accuracy) from the food-related Stroop task. HCF-C, high-calorie foods in congruent trials; HCF-INC, high-calorie foods in incongruent trials; LCF-C, low-calorie foods in congruent trials; LCF-INC, low-calorie foods in incongruent trials. * *p* < 0.05; *** *p* < 0.001.

**Figure 3 nutrients-14-04593-f003:**
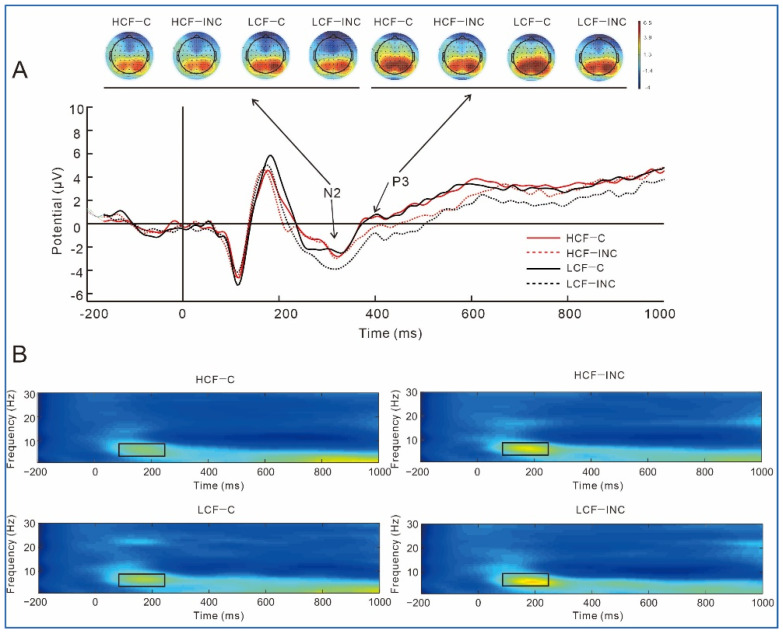
(**A**). Stimuli-locked, grand average waveforms of N2 and P3 at site Fz; (**B**). The theta power at site Fz. HCF−C, high-calorie foods in congruent trials; HCF−INC, high-calorie foods in incongruent trials; LCF−C, low-calorie foods in congruent trials; LCF−INC, low-calorie foods in incongruent trials.

## Data Availability

The data presented in this study are available on request from the corresponding author.

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
