# Peer review of "High-Calorie Food-Cues Impair Conflict Control: EEG Evidence from a Food-Related Stroop Task"

_nutrients, 2022, doi:10.3390/nu14214593_

Round 1

Reviewer 1 Report

The article is interesting but extremely difficult to understand. The text contains mixtures of ideas and technical details. These must be separated. Also the text has question marks in the middle of a sentence and half sentences, which must be changed.

My recommendation is that the text, already from the Inroduction must be made clear and not technical. I am convinced the information is important. First of all, the conflict control must me defined. What is meant by that?

Author Response

The article is interesting but extremely difficult to understand. The text contains mixtures of ideas and technical details. These must be separated. Also the text has question marks in the middle of a sentence and half sentences, which must be changed.

Response:Thank you for your suggestions for improving our manuscript. In the Introduction section of the manuscript, we firstly introduced the behavioral data of previous studies on conflict control, and then introduced the existing ERP evidence. The Introduction section has been reconstructed for a clearer delivery. Before introducing the corresponding information mentioned above, some additional comments about the existing literature were made in the revised manuscript.

 Additionally, we have revised the spelling errors (question marks).

My recommendation is that the text, already from the Inroduction must be made clear and not technical. I am convinced the information is important. First of all, the conflict control must me defined. What is meant by that?

Response:Thank you for your suggestions.

ERP is a neuroscience tool with high temporal resolution and is often used in cognitive psychology research. The changes in ERPs can indicate the changes in conflict control. The findings using ERP cannot be separated since they should be viewed with its entirety. However, other key background information provided in the original manuscript has been reorganized to form a more consistent structure. In the Introduction section of the manuscript, we firstly introduced the studies on conflict control at the level of behavior, and then introduced the ERP evidences. The above changes have been made to the revised manuscript.

 Conflict control refers to the information processing capability of the brain to monitor and respond to the emergence of conflicts, and is considered a core component of the brain’s executive function (see, P1 L44-46)

Reviewer 2 Report

While the concept of linking cognitive responses to brain potentials is of considerable interest the experimental methods need to be further characterized to eliminate the other aspects of these stimuli (different colors, different food groups, different nutrient density) needs to be shown before the conclusions reached by the authors can be considered valid.

In addition, the manuscript has major difficulties with the English language which makes it difficult to understand and follow.  A native speaker of English should review and correct the revised manuscript before it is resubmitted.

There appears to be a missing line between lines 13 and 14 in the abstract it currently reads “                   A food-related Stroop task employed.”  Which is not a sentence and makes no sense.

The introduction is unclear and often the references do not support the statements made.  There needs to be a revision using outside reference sources not just the work of this laboratory.

Line 29 “constantly struggling between health food and unhealthy food temptations”  is “healthy” intended?  What is the scientific backing for this statement ref 1 discusses restraint and calorie dense foods.  The way the authors are phrasing this introduces bias (seen throughout the introduction). What constitutes “health food” and unhealthy food is a topic of considerable debate with such foods as meats, fruits, milk, etc all being the subject of debate.

Lines 32-34 References are all self references and are not sufficient to substantiate the claim being made.

Line 39:  conflict control does not equal healthy – a more precise definition and explanation is needed

Line 47 why is there a question mark?

Line 97 (this type of English language error is seen throughout) …foods on conflict control among healthy populations using food-related Stroop task”  should be either “..foods on conflict control among healthy populations using a food-related Stroop task”  or “…foods on conflict control among healthy populations using food-related Stroop tasks”.

Lines 109 ff:  it states that: subjects “were required to abstain from taking substances or medications that could potentially influence their concentrations before participating in the study; they were also required  to disclose any history of major psychological disorders.

Did these substances include tea (which contains caffeine) was there any monitoring of caffeine intake?

The methodology description needs to be expanded especially concerning the  choice of food stimuli.  This also needs to be addressed in the introduction.  The stimuli not only differ in terms of calories (discussed in the paper) but also in nutrient density.  This needs to be discussed as it is a potential confound. The validity and reliability of the chosen food based stimuli also needs to be addressed. 

This is also the major problem with the discussion as it focuses on the differences in caloric content when the chosen stimuli differ in many ways including color which is the basis for many Stoop type tasks.  The specific stimuli need to be better characterized and the associations with these aspects of the stimuli should be part of the discussion and conclusion.

Author Response

While the concept of linking cognitive responses to brain potentials is of considerable interest the experimental methods need to be further characterized to eliminate the other aspects of these stimuli (different colors, different food groups, different nutrient density) needs to be shown before the conclusions reached by the authors can be considered valid.

Response: Thank you for your suggestions. The stimuli (high-calorie and low-calorie foods) used in the study were from our previous studies. However, we did not consider the influence of other characteristics such as the stimuli colors on the experimental results, which is a limitation of the study. We have added the limitation to the discussion.  (see, P10 L459-461; P11 L462-464)

In addition, the manuscript has major difficulties with the English language which makes it difficult to understand and follow.  A native speaker of English should review and correct the revised manuscript before it is resubmitted.

Response: The manuscript was reviewed by a Native-English-speaking professional academic editor before resubmission.

There appears to be a missing line between lines 13 and 14 in the abstract it currently reads “  A food-related Stroop task employed.”  Which is not a sentence and makes no sense.

Response: Thank you for your suggestions. We have revised the sentence in the revised manuscript. 

The introduction is unclear and often the references do not support the statements made.  There needs to be a revision using outside reference sources not just the work of this laboratory.

Response: Thank you for your suggestions. We have added and deleted some references. In addition, we rearranged the Introduction section.

 Line 29 “constantly struggling between health food and unhealthy food temptations”  is “healthy” intended?  What is the scientific backing for this statement ref 1 discusses restraint and calorie dense foods.  The way the authors are phrasing this introduces bias (seen throughout the introduction). What constitutes “health food” and unhealthy food is a topic of considerable debate with such foods as meats, fruits, milk, etc all being the subject of debate.

Response: Thank you for your suggestions. Based on previous studies (see below), high-calorie foods have been considered as unhealthy foods while low-calorie foods have been considered as healthy foods. Therefore, we described high calorie foods as unhealthy foods in the current study.

Calitri, R., Pothos, E. M., Tapper, K., Brunstrom, J. M., & Rogers, P. J. (2010). Cognitive biases to healthy and unhealthy food words predict change in BMI. Obesity18(12), 2282-2287.

Cosme, D., Zeithamova, D., Stice, E., & Berkman, E. T. (2020). Multivariate neural signatures for health neuroscience: assessing spontaneous regulation during food choice. Social Cognitive and Affective Neuroscience15(10), 1120-1134.

Smith, R., Alkozei, A., & Killgore, W. D. (2018). Conflict-related dorsomedial frontal cortex activation during healthy food decisions is associated with increased cravings for high-fat foods. Brain Imaging and Behavior12(3), 685-696.

Veling, H., Aarts, H., & Stroebe, W. (2013). Using stop signals to reduce impulsive choices for palatable unhealthy foods. British journal of health psychology18(2), 354-368.

Lines 32-34 References are all self references and are not sufficient to substantiate the claim being made.

Response: Thank you for your suggestions. We have added some corresponding references in the revise manuscript.

Line 39:  conflict control does not equal healthy – a more precise definition and explanation is needed.

Response: Thank you for your suggestions. Conflict control refers to the information processing capability of the brain to monitor and respond to the emergence of conflicts, and is considered a core component of the brain’s executive function (see, P1 L44-46)

Line 47 why is there a question mark?

Response: This is a spelling error and has been corrected in the revised manuscript.

Line 97 (this type of English language error is seen throughout) …foods on conflict control among healthy populations using food-related Stroop task”  should be either “..foods on conflict control among healthy populations using a food-related Stroop task”  or “…foods on conflict control among healthy populations using food-related Stroop tasks”.

Response: Thank you for your suggestions. We have revised the English language error. In addition, the manuscript was reviewed by a Native-English-speaking professional academic editor before resubmission. 

Lines 109 ff:  it states that: subjects “were required to abstain from taking substances or medications that could potentially influence their concentrations before participating in the study; they were also required to disclose any history of major psychological disorders. Did these substances include tea (which contains caffeine) was there any monitoring of caffeine intake?

Response: These substances included tea and coffee. We have added the corresponding information in the revised manuscript. (see, P4 L182)

The methodology description needs to be expanded especially concerning the choice of food stimuli.  This also needs to be addressed in the introduction.  The stimuli not only differ in terms of calories (discussed in the paper) but also in nutrient density.  This needs to be discussed as it is a potential confound. The validity and reliability of the chosen food based stimuli also needs to be addressed. This is also the major problem with the discussion as it focuses on the differences in caloric content when the chosen stimuli differ in many ways including color which is the basis for many Stoop type tasks.  The specific stimuli need to be better characterized and the associations with these aspects of the stimuli should be part of the discussion and conclusion.

Response: Thank you for your suggestions. These are very important and constructive suggestions. The stimuli used in the present were from our food-picture lab of our team. We only focus on the caloric information of the stimuli and classify food into high-calorie and low-calorie food. However, we did not consider the influence of other characteristics such as the stimuli colors or nutrient density on the experimental results, which is a limitation of the study. We have added the limitation to the discussion. (see, P10 L459-461; P11 L462-464)

Round 2

Reviewer 2 Report

While there are still some minor editing issues the manuscript is greatly improved.  The introduction is now much improved and explains the issues to a greater extent.  Would suggest minor review of the discussion to better focus on the content of the introduction.

Author Response

While there are still some minor editing issues the manuscript is greatly improved.  The introduction is now much improved and explains the issues to a greater extent.  Would suggest minor review of the discussion to better focus on the content of the introduction.

Response: Thank you for your suggestions. We have reconstructed Discussion based on the content of the Introduction.
